# Evaluating the Knowledge Base Completion Potential of GPT

**Blerta Veseli[1], Simon Razniewski[2], Jan-Christoph Kalo[3], Gerhard Weikum[1]**

Max Planck Institute for Informatics[1], Bosch Center for AI[2], University of Amsterdam[3]

{weikum,bveseli}@mpi-inf.mpg.de

simon.razniewski@de.bosch.com

j.c.kalo@uva.nl

## Abstract

Structured knowledge bases (KBs) are an asset for search engines and other applications, but are inevitably incomplete. Language models (LMs) have been proposed for unsupervised knowledge base completion (KBC), yet, their ability to do this at scale and with high accuracy remains an open question. Prior experimental studies mostly fall short because they only evaluate on popular subjects, or sample already existing facts from KBs. In this work, we perform a careful evaluation of GPT's potential to *complete* the largest public KB: Wikidata. We find that, despite their size and capabilities, models like GPT-3, ChatGPT and GPT-4 do not achieve fully convincing results on this task. Nonetheless, they provide solid improvements over earlier approaches with smaller LMs. In particular, we show that, with proper thresholding, GPT-3 enables to extend Wikidata by 27M facts at 90% precision.

## 1 Introduction

Structured knowledge bases (KBs) like Wikidata (Vrandečić and Krötzsch, 2014), DBpedia (Auer et al., 2007), and Yago (Suchanek et al., 2007) are employed in many knowledge-centric applications like search, question answering and dialogue. Constructing and completing these KBs at high quality and scale is a long-standing research challenge, and multiple benchmarks exist, e.g., FB15k (Bordes et al., 2013), CoDEx (Safavi and Koutra, 2020), and LM-KBC22 (Singhania et al., 2022). Text-extraction, knowledge graph embeddings, and LM-based knowledge extraction have continuously moved scores upwards on these tasks, and leaderboard portals like Paperswithcode[1] provide evidence for that.

Recently, LMs have been purported as a promising source of structured knowledge. Starting from

the seminal LAMA paper (Petroni et al., 2019), a throve of works have explored how to better probe, train, or fine-tune these LMs (Liu et al., 2022).

Nonetheless, we observe a certain divide between these late-breaking investigations, and *practical KB completion*. While recent LM-based approaches often focus on simple methodologies that produce fast results, practical KBC so far is a highly precision-oriented, extremely laborious process, involving a very high degree of manual labour, either for manually creating statements (Vrandečić and Krötzsch, 2014), or for building comprehensive scraping, cleaning, validation, and normalization pipelines (Auer et al., 2007; Suchanek et al., 2007). For example, part of Yago's success stems from its validated >95% accuracy, and according to (Weikum et al., 2021), the Google Knowledge Vault was not deployed into production partly because it did not achieve 99% accuracy. Yet, many previous LM analyses balance precision and recall or report precision/hits@k values, implicitly tuning systems towards balanced recall scores resulting in impractical precision. It is also important to keep in mind the scale of KBs: Wikidata currently contains around 100 million entities and 1.2B statements. The cost of producing such KBs is massive. An estimate from 2018 sets the cost per statement at 2 $ for manually curated statement, and 1 ct for automatically extracted ones (Paulheim, 2018). Thus, even small additions in relative terms might correspond to massive gains in absolute numbers. For example, even by the lower estimate of 1 ct/statement, adding one statement to just 1% of Wikidata humans would come at a cost of 100,000 $.

In this paper, *we conduct a systematic analysis of the KB completion potential of GPT,* where we focus on *high precision*. We evaluate by employing (i) a recent KB completion benchmark, WD-KNOWN, (Veseli et al., 2023), which randomly samples facts from Wikidata and (ii) by a manual evaluation of subject-relation pairs without object

---

[1] https://paperswithcode.com/task/knowledge-graph-completion

values. Our main results are:

1. For the long-tail entities of WD-KNOWN, GPT models perform considerably worse than what less demanding benchmarks like LAMA (Petroni et al., 2019) have indicated. Nonetheless, we can achieve solid results for language-related, socio-demographic relations (e.g., *nativeLanguage*).

2. Despite their fame and size, out of the box, the GPT models, including GPT-4, do not produce statements of a high enough accuracy as typically required for KB completion.

3. With simple thresholding, for the first time, we obtain a method that can extend the Wikidata KB at extremely high quality (>90% precision), at the scale of millions of statements. Based on our analysis of 41 common relations, we would be able to add a total of 27M high-accuracy statements.

## 2  Background and Related Work

**KB construction**   KB construction has a considerable history. One prominent approach is by human curation, as done e.g., in the seminal Cyc project (Lenat, 1995), and this is also the backbone of today's most prominent public KB, Wikidata (Vrandečić and Krötzsch, 2014). Another popular paradigm is the extraction from semi-structured resources, as pursued in Yago and DBpedia (Suchanek et al., 2007; Auer et al., 2007). Extraction from free text has also been explored (e.g., NELL (Carlson et al., 2010)). A popular paradigm has been embedding-based link prediction, e.g., via tensor factorization like Rescal (Nickel et al., 2011), and KG embeddings like TransE (Bordes et al., 2013).

An inherent design decision in KBC is the P/R trade-off – academic projects are often open to trade these freely (e.g., via F-1 scores), yet production environments are often critically concerned with precision, e.g., Wikidata generally discouraging statistical inferences, and industrial players likely use to a considerable degree human editing and verification (Weikum et al., 2021).

For example in all of Rescal, TransE, and LAMA, the main results focus on metrics like hits@k, MRR, or AUC, which provide no bounds on precision.

**LMs for KB construction**   Knowledge extraction from LMs provides fresh hope for the synergy of automated approaches and high-precision curated KBs. It provides remarkably straightforward access to very large text corpora: The basic idea by (Petroni et al., 2019) is to just define one template per relation, then query the LM with subject-instantiated versions, and retain its top prediction(s). A range of follow-up works appeared, focusing, e.g., on investigating entities, improving updates, exploring storage limits, incorporating unique entity identifiers, and others (Shin et al., 2020; Poerner et al., 2020; Cao et al., 2021; Roberts et al., 2020; Heinzerling and Inui, 2021; Petroni et al., 2020; Elazar et al., 2021; Razniewski et al., 2021; Cohen et al., 2023; Sun et al., 2023). Nonetheless, we observe the same gaps as above: The high-precision area, and completion of already existing resources, are not well investigated.

Several works have analyzed the potential of larger LMs, specifically GPT-3 and GPT-4,. They investigate few-shot prompting for extracting factual knowledge for KBC (Alivanistos et al., 2023) or for making the factual knowledge in a LM more explicit (Cohen et al., 2023). These models can aid in building a knowledge base on Wikidata or improving the interpretability of LMs. Despite the variance in the precision of extracted facts from GPT-3, it can peak at over 90% for some relations.

Recently, GPT-4's capabilities for KBC and reasoning were examined (Zhu et al., 2023). This research compared GPT-3, ChatGPT, and GPT-4 on information extraction tasks, KBC, and KG-based question answering. However, these studies focus on popular statements from existing KBs, neglecting the challenge of introducing genuinely new knowledge in the long tail.

In (Veseli et al., 2023), we analyzed to which degree BERT can complete the Wikidata KB, i.e., provide novel statements. Together with the focus on high precision, this is also the main difference of the present work to the works cited above, which evaluate on knowledge already existing in the KB, and do not estimate how much they could add.

## 3  Analysis Method

**Dataset**   We consider the 41 relations from the LAMA paper (Petroni et al., 2019). For automated evaluation and threshold finding, we employ the WD-KNOWN dataset (Veseli et al., 2023). Unlike other KBC datasets, this one contains truly long-

| Relation | GPT-4 | | | GPT-3 text-davinci-003 | | | ChatGPT gpt-3.5-turbo | | |
|---|---|---|---|---|---|---|---|---|---|
| | P | R | F1 | P | R | F1 | P | R | F1 |
| writtenIn | 0.62 | 0.59 | 0.6 | 0.91 | 0.78 | 0.84 | 0.37 | 0.85 | 0.52 |
| ownedBy | 0.41 | 0.3 | 0.35 | 0.6 | 0.44 | 0.51 | 0.17 | 0.61 | 0.27 |
| nativeLanguage | 0.85 | 0.85 | 0.85 | 0.69 | 0.86 | 0.77 | 0.53 | 0.88 | 0.66 |
| LanguageOfFilm | 0.78 | 0.63 | 0.7 | 0.58 | 0.52 | 0.55 | 0.48 | 0.65 | 0.55 |
| hasCapital | 0.77 | 0.48 | 0.59 | 0.77 | 0.44 | 0.56 | 0.49 | 0.86 | 0.62 |
| officialLanguage | 0.62 | 0.6 | 0.61 | 0.67 | 0.64 | 0.65 | 0.27 | 0.73 | 0.39 |
| foundedIn | 0.27 | 0.53 | 0.36 | 0.4 | 0.38 | 0.39 | 0.14 | 0.58 | 0.23 |
| playsInstrument | 0.25 | 0.36 | 0.3 | 0.16 | 0.18 | 0.17 | 0.13 | 0.6 | 0.21 |
| partOf | 0.05 | 0.1 | 0.06 | 0.17 | 0.1 | 0.13 | 0.06 | 0.36 | 0.1 |
| citizenOf | 0.72 | 0.62 | 0.67 | 0.67 | 0.6 | 0.63 | 0.47 | 0.68 | 0.56 |
| spokenLanguage | 0.48 | 0.62 | 0.54 | 0.54 | 0.76 | 0.63 | 0.37 | 0.84 | 0.51 |
| playerPosition | 0.4 | 0.4 | 0.4 | 0.18 | 0.24 | 0.21 | 0.23 | 0.7 | 0.35 |
| inContinent | 0.62 | 0.56 | 0.59 | 0.61 | 0.6 | 0.6 | 0.4 | 0.66 | 0.5 |
| namedAfter | 0.5 | 0.49 | 0.49 | 0.53 | 0.44 | 0.48 | 0.12 | 0.36 | 0.18 |
| hostCountry | 0.77 | 0.5 | 0.61 | 0.75 | 0.48 | 0.59 | 0.4 | 0.55 | 0.46 |
| musicLabel | 0.29 | 0.31 | 0.3 | 0.16 | 0.18 | 0.17 | 0.08 | 0.48 | 0.14 |
| hasReligion | 0.44 | 0.36 | 0.4 | 0.47 | 0.38 | 0.42 | 0.16 | 0.39 | 0.23 |
| developedBy | 0.43 | 0.7 | 0.53 | 0.45 | 0.5 | 0.47 | 0.11 | 0.6 | 0.19 |
| countryOfJurisdiction | 0.38 | 0.24 | 0.29 | 0.52 | 0.22 | 0.4 | 0.42 | 0.33 | 0.37 |
| subclassOf | 0.21 | 0.4 | 0.28 | 0.73 | 0.85 | 0.79 | 0.16 | 0.62 | 0.25 |
| diplomaticRelation | 0.36 | 0.62 | 0.46 | 0.7 | 0.17 | 0.27 | 0.41 | 0.32 | 0.36 |
| CountryOfOrigin | 0.6 | 0.34 | 0.43 | 0.48 | 0.31 | 0.38 | 0.2 | 0.39 | 0.26 |
| Macro-Average | 0.49 | 0.48 | 0.47 | 0.53 | 0.46 | 0.48 | 0.28 | 0.59 | 0.36 |

Table 1: Automated evaluation in the *retain-all setting*: GPT-3 (text-davinci-003 with 175B parameters), GPT-4 (#parameters unknown) and ChatGPT (gpt-3.5-turbo with #parameters unknown ) on 1000 samples/relation from WD-Known.

tail entities, by randomly sampling from Wikidata, a total of 4 million statements for 3 million subjects in 41 relations (Petroni et al., 2019). Besides this dataset for automated evaluation, for the main results, we use manual evaluation on Wikidata entities that *do not yet have the relations of interest*. For this purpose, for each relation, we manually define a set of relevant subject types (e.g., software for developedBy), that allows us to query for subjects that miss a property.

**Evaluation protocol** In the automated setting, we first use a *retain-all* setting, where we evaluate the most prominent GPT models (GPT-3 text-davinci-003, GPT-4, and ChatGPT gpt-3.5-turbo) by precision, recall, and F1. Table 1 shows that none of the GPT models could achieve precision of >90%. In a second step, the *precision-thresholding* setting, we therefore sort predictions by confidence and evaluate by recall at precision 95% and 90% (R@P95 and R@P90). To do so, we sort the predictions for all subjects in a relation by the model's probability on the first generated token[2], then compute the precision at each point of this list, and return the maximal fraction of the list covered while maintaining precision greater than the de-

---

[2]This is a heuristic only, as unbiased probabilities are not easy to assign to multi-token generations in list answers (Singhania et al., 2023).

sired value. We threshold only GPT-3, because only GPT-3's token probabilities are directly accessable in the API, and because the chat-aligned models do not outperform it in the retain-all setting. Approaches to estimate probabilities post-hoc can be found in (Xiong et al., 2023).

Since automated evaluations are only possible for statements already in the KB, in a second step, we let human annotators evaluate the correctness of 800 samples of *novel* (out-of-KB) high-accuracy predictions. We hereby use a relation-specific threshold determined from the automated 75%-95% precision range. MTurk annotators could use Web search to verify the correctness of our predictions on a 5-point Likert scale (correct/likely/unknown/implausible/false). We counted predictions that were rated as correct or likely as true predictions, and all others as false.

**Prompting setup** To query the GPT models, we utilize instruction-free prompts listed in the appendix. Specifically for GPT-3, we follow the prompt setup of (Cohen et al., 2023), which is based on an instruction-free prompt entirely consisting of 8 randomly sampled and manually checked examples. In the default setting, all example subjects have at least one object. Since none of the GPT models achieved precision >90% and we can only threshold GPT-3 for high precision, we focus on the largest GPT-3 model (text-davinci-003) in the following. We experimented with three variations for prompting this model:

1. **Examples w/o answer**: Following (Cohen et al., 2023), in this variant, we manually selected 50% few-shot examples, where GPT-3 did not know the correct answer, to teach the model to output "Don't know". This is supposed to make the model more conservative in cases of uncertainty.

2. **Textual context augmentation**: Following (Petroni et al., 2020), we test whether adding textual context improves model performance. We hereby employ Google Web Search, with the subject and relation of interest as search query. The top 1 result snippet is then included as context to the prompt.

3. **#few-shot examples**: A standard parameter in prompting is the number of few-shot examples. They have a huge impact on monetary costs. We vary this number between 1 and 12.

| Relation | #current stmts. in Wikidata | #subj. w/ missing stmts. | fraction for which GPT-3 can give high-confidence prediction | #addable stmts. | manual accuracy | relative growth |
|---|---|---|---|---|---|---|
| foundedIn | 43,254 | 225,578 | 9% | 20,302 | **92%** | 43% |
| citizenOf | 4,206,684 | 4,616,601 | 5% | 230,830 | 82% | 5% |
| countryOfJurisdiction | 901,066 | 24,966 | 76% | 18,974 | 88% | 2% |
| namedAfter | 340,234 | 477,845 | 22% | 105,125 | 64% | 20% |
| inContinent | 71,101 | 889,134 | 62% | 551,263 | 88% | 682% |
| ownedBy | 449,140 | 416,437 | 6% | 24,986 | 24% | 1% |
| hostCountry | 14,275,596 | 35,214 | 53% | 18,663 | 88% | 0% |
| spokenLanguage | 2,148,775 | 7,134,543 | 57% | 4,066,689 | **92%** | 174% |
| writtenIn | 14,140,453 | 24,990,161 | 73% | 18,242,817 | **92%** | 119% |
| officialLanguage | 19,678 | 6,776 | 42% | 2,846 | **100%** | 14% |
| developedBy | 42,379 | 29,349 | 6% | 1,761 | **94%** | 4% |
| CountryOfOrigin | 1,296,038 | 135,196 | 49% | 66,246 | 30% | 2% |
| hasCapital | 111,171 | 973 | 11% | 107 | 14% | 0% |
| LanguageOfFilm | 337,682 | 70,669 | 24% | 16,961 | 82% | 4% |
| nativeLanguage | 264,778 | 7,871,085 | 49% | 3,856,831 | 82% | 1195% |
| sharesBorders | 6,946 | 222 | 14% | 31 | 72% | 0% |
| Overall | 38,654,975 | 46,924,749 | 66% | 27,224,432 | 90 % | 70% |

Table 2: Manual evaluation: Wikidata KB completion potential of GPT-3 text-davinci-003 with precision-oriented thresholding.

## 4 Results and Discussion

**Can GPT models complete Wikidata at precision AND scale?** In Table 1 we already showed that without thresholding, none of the GPT models can achieve sufficient precision. Table 2 shows our main results when using precision-oriented thresholding, on the 16 best-performing relations. The fourth column shows the percentage of subjects for which we obtained high-confidence predictions, the fifth how these translates into absolute statement numbers, and the sixth shows the percentages that were manually verified as correct (sampled). In the last column, we show how this number relates to the current size of the relation.

We find that manual precision surpasses 90% for 5 relations, and 80% for 11. Notably, the best-performing relations are mostly related to socio-demographic properties (languages, citizenship).

In absolute terms, we find a massive number of high-accuracy statements that could be added to the *writtenIn* relation (18M), followed by *spokenLanguage* and *nativeLanguage* (4M each). In relative terms, the additions could increase the existing relations by up to 1200%, though there is a surprising divergence (4 relations over 100%, 11 relations below 20%).

**Does GPT provide a quantum leap?** Generating millions of novel high-precision facts is a significant achievement, though the manually verified precision is still below what industrial KBs aim for. The wide variance in relative gains also shows that GPT only shines in selected areas. In line with previous results (Veseli et al., 2023), we find that GPT can do well on relations that exhibit high surface correlations (person names often give away their nationality), otherwise the task remains hard.

In Table 3 we report the automated evaluation of precision-oriented thresholding. We find that on many relations, GPT-3 can reproduce existing statements at over 95% precision, and there are significant gains over the smaller BERT-large model. At the same time, it should be noted that (Sun et al., 2023) observed that for large enough models, parameter scaling does not improve performance further, so it is well possible that these scores represent a ceiling w.r.t. model size.

**Is this cost-effective?** Previous works have estimated the cost of KB statement construction at 1 ct. (highly automated infobox scraping) to $2 (manual curation) (Paulheim, 2018). Based on our prompt size (avg. 174 tokens), the cost of one query is about 0.35 ct., with filtering increasing the cost per retained statement to about 0.7 ct. So LM prompting is monetarily competitive to previous infobox scraping works, though with much higher recall potential.

In absolute terms, prompting GPT-3 for all 48M incomplete subject-relation pairs reported in Table 2 would amount to an expense of $168,000, and yield approximately 27M novel statements.

**Does "Don't know" prompting help?** In Table 4 (middle) we show the impact of using examples without an answer. The result is unsystematic, with notable gains in several relations, but some losses in others. Further research on calibrating model confidences seems important (Jiang et al., 2021; Singhania et al., 2023).

| | GPT-3 text-davinci-003 | | GPT-3 text-curie-001 | | BERT$_{Large}$ | |
|---|---|---|---|---|---|---|
| Relation | $R@P95$ | $R@P90$ | $R@P95$ | $R@P90$ | $R@P95$ | $R@P90$ |
| writtenIn | 0.69 | 0.76 | 0.2 | 0.39 | 0 | 0 |
| ownedBy | 0.37 | 0.39 | 0.11 | 0.16 | 0 | 0 |
| nativeLanguage | 0.22 | 0.7 | 0.11 | 0.6 | 0.43 | 0.58 |
| LanguageOfFilm | 0.21 | 0.33 | 0 | 0.01 | 0 | 0.01 |
| hasCapital | 0.19 | 0.31 | 0 | 0 | 0.02 | 0.02 |
| officialLanguage | 0.09 | 0.24 | 0 | 0 | 0.03 | 0.25 |
| foundedIn | 0.06 | 0.08 | 0 | 0 | 0 | 0 |
| playsInstrument | 0.02 | 0.02 | 0 | 0 | 0 | 0 |
| partOf | 0.01 | 0.01 | 0 | 0 | 0 | 0 |
| citizenOf | 0.01 | 0.24 | 0 | 0 | 0.02 | 0.03 |
| spokenLanguage | 0.01 | 0.47 | 0.02 | 0.13 | 0 | 0.24 |
| playerPosition | 0.01 | 0.02 | 0 | 0 | 0 | 0 |
| inContinent | 0.01 | 0.01 | 0 | 0 | 0 | 0 |
| namedAfter | 0.01 | 0.07 | 0.01 | 0.01 | 0 | 0 |
| hostCountry | 0.01 | 0.34 | 0.02 | 0.02 | 0 | 0 |
| musicLabel | 0.01 | 0.02 | 0 | 0 | 0 | 0 |
| hasReligion | 0 | 0.02 | 0 | 0 | 0.03 | 0.07 |
| developedBy | 0 | 0.11 | 0 | 0 | 0.04 | 0.04 |
| countryOfJurisdiction | 0 | 0.08 | 0.05 | 0.08 | 0.02 | 0.02 |
| subclassOf | 0 | 0.37 | 0 | 0 | 0 | 0 |
| diplomaticRelation | 0 | 0.01 | 0 | 0 | 0 | 0 |
| CountryOfOrigin | 0 | 0.01 | 0 | 0 | 0.01 | 0.01 |

Table 3: Automated evaluation in the *precision-thresholding* setting: GPT-3 (text-davinci-003 with 175B parameters, text-curie-001 with 6.7B parameters) and BERT-large (340M parameters) on 1000 samples/relation from WD-Known.

| | Standard | | Don't know | | Standard and textual context | |
|---|---|---|---|---|---|---|
| Relation | $R@P95$ | $R@P90$ | $R@P95$ | $R@P90$ | $R@P95$ | $R@P90$ |
| nativeLanguage | 0.22 | **0.7** | **0.56** | 0.68 | 0 | 0 |
| foundedIn | 0.06 | 0.08 | **0.09** | **0.13** | 0 | 0.01 |
| developedBy | 0 | 0.11 | **0.06** | **0.18** | 0 | 0 |
| spokenLanguage | 0.01 | **0.47** | 0.01 | 0.04 | 0 | 0 |
| employedBy | 0 | 0 | **0.01** | **0.01** | 0 | 0 |
| inContinent | **0.01** | **0.01** | 0 | 0 | 0 | 0 |
| citizenOf | 0.01 | 0.24 | 0 | 0.24 | **0.03** | 0.04 |

Table 4: Effect of variations to the standard prompting setting.

**Does textual context help?** Table 4 (right) shows the results for prompting with context. Surprisingly, this consistently made performance worse, with hardly any recall beyond 90% precision. This is contrary to earlier findings like (Petroni et al., 2020) (for BERT) or (Mallen et al., 2023) (for QA), who found that context helps, especially in the long tail. Our analysis indicates that, in the high-precision bracket, misleading contexts cause more damage (lead to high confidence in incorrect answers), than what helpful contexts do good (boost correct answers).

**How many few-shot examples should one use?** Few-shot learning for KBC works with remarkably few examples. While our default experiments, following (Cohen et al., 2023), used 8 examples, we found actually no substantial difference to smaller example sizes as low as 4.

## 5 Conclusion

We provided the first analysis of the real KB completion potential of GPT. Our findings indicate that GPT-3 could add *novel* knowledge to Wikidata, at unprecedented scale and quality (27M statements at 90% precision). Compared with other approaches the estimated cost of $168,000 is surprisingly cheap, and well within the reach of industrial players. We also find that, in the high-precision bracket, GPT-3 distills web content to a degree that context augmentation does not easily help.

Open issues remain in particular around identifying high-confidence predictions within an LM's generations (Jiang et al., 2021; Singhania et al., 2023; Xiong et al., 2023), and the choice of examples.

## Limitations

Using LMs for automated knowledge generation comes with the standard risk of exacerbating demographic biases. For example, many of the best-performing relations are language-related, where the model presumably often estimates a person's native language entirely from their name.

In terms of reproducibility, it should be noted that our results are tied to a closed-source commercial API. Although GPT-3/4/chatGPT are widely used in research and industry, and OpenAI has announced plans to keep stable model versions online, long-term reproducibility is not ensured, and the internal workings of GPT are publicly not known.

Although statement generation is at the core of KB completion, for a complete KBC pipeline, we are still missing critical components. This concerns in particular entity disambiguation, which is essential for relations with more than a few hundred possible object values. Similarly, Wikidata and other KBs give critical importance to scrutable referencing of statements. This is not easily possible with LMs.

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

## A Predictions

In the following we show some predictions from our different experimental setups. Q marks the query, P the prediction by GPT-3 and A the ground truth object. Please note that to prompt GPT-3, relation-specific examples are used at the beginning of each query. The prompts w/ examples can be seen in Appendix B.

### A.1 Standard

This section shows predictions from our standard setup with the number of few-shot examples being 8. The prompts B.1.

### A.1.1 GPT-3

**Q:** As It Is in Heaven # original language
**P:** Swedish
**A:** Swedish

**Q:** John Gruber # employer
**P:** Daring Fireball
**A:** The Pittsburg Times, Pittsburgh Post, Pittsburgh Gazette, The Gazette Times, Pittsburgh Pirates

**Q:** Hillary Canyon # continent
**P:** Antarctica
**A:** North America

**Q:** The Incredible Hulk # developer
**P:** Edge of Reality
**A:** Edge of Reality, Amaze Entertainment

**Q:** Architecture and secondary structure of an entire HIV-1 RNA genome # language of work
**P:** English
**A:** English

### A.1.2 GPT-4

**Q:** Who are the developers of The Incredible Hulk? Give me a list with no additional text.
**P:** Edge of Reality
**A:** Radical Entertainment, Edge of Reality, Amaze Entertainment, Pocket Studios, Universal Interactive

**Q:** In which languages is Architecture and secondary structure of an entire HIV-1 RNA genome available? Give me a list with no additional text.
**P:** English
**A:** I'm sorry, but I couldn't find specific information about the languages in which the mentioned scientific study or information is available.

### A.1.3 ChatGPT

**Q:** Who are the developers of The Incredible Hulk? Give me a list with no additional text.
**P:** Edge of Reality
**A:** Edge of Reality, Amaze Entertainment, Beenox, Foundation 9 Entertainment, Griptonite Games, Fizz Factor, Nucore

**Q:** In which languages is In which languages is Architecture and secondary structure of an entire HIV-1 RNA genome available? Give me a list with no additional text. available? Give me a list with no additional text.
**P:** English
**A:** English

**A:** Kingdom of Italy, Italy

## A.2 Textual context

In this section we show some predictions from our standard setup with textual contexts. The examples used for prompting in this setup can be seen in section B.2.

**Q:** Neko-Nin exHeart # developer
**C:** Apr 21, 2017 ... Developer.
Whirlpool ; Publisher. Sekai
Project ; Released. Apr 21, 2017
; OS: Windows 7 or above ; Pro-
cessor: 1.2 GHz Pentium 4
**P:** Whirlpool, Sekai Project
**A:** Whirlpool

**Q:** Navel*Plus # developer
**C:** Position: Qlik Sense Devel-
oper. Location: Phoenix, AZ ...
SSIS and Datastage are a plus.
Thanks & Regards,. Sravan Kumar
| Navel Technologies Inc.
**P:** Navel Technologies Inc.
**A:** Navel

**Q:** Andrei Krasilnikov # native
language
**C:** Languages · Russian. Native
or bilingual proficiency · En-
glish. Full professional profi-
ciency · French. Limited working
proficiency · Spanish. Limited
working
**P:** Spanish, French, English, Rus-
sian
**A:** Russian

**Q:** Sergio Aquilante # citizen-
ship
**C:** View Ariana Morais' profile
on LinkedIn, the world's largest
professional community. Ariana
has 1 job listed on their profile.
See the complete profile on
**P:** Brazil

**Q:** Luis Alfredo López # citizen-
ship
**C:** Congratulations to Luis "Al-
fredo" Lopez, an associate from
Vistar - Southern California, for
being inducted into the 2020 IFDA
Truck Driver Hall of Fame
**P:** Mexico
**A:** Honduras

## A.3 Don't Know

In this section we present some predictions from our setup using Don't know prompting. The examples used for prompting in this setup can be seen in section B.3.

**Q:** Visvanath Kar # native lan-
guage
**P:** Don't know
**A:** Odia

**Q:** Eikeviken # manufacturer
**P:** Don't know
**A:** Samsung Heavy Industries

**Q:** Volkwin Marg # work location
**P:** Hamburg, Aachen
**A:** Berlin, Brunswick, Hamburg,
Aachen

**Q:** Elvive # manufacturer
**P:** L'Oréal
**A:** L'Oréal

## A.4 Out-of-KB facts

In this section we present some predictions for out-of-KB facts evaluated in a manual evaluation. For prompting we used the examples from our standard setup B.1.

**Q:** Lalith Perera # citizenship
**P:** Sri Lanka

**Q:** The Splatters # developer
**P:** SpikySnail Games

**Q:** Perú Negro # location of formation
**P:** Lima

**Q:** Albrecht von Quadt # languages spoken, written or signed in
**P:** German

**Q:** Wreckless: The Yakuza Missions # developer
**P:** Bunkasha Games

## B Prompts

### B.1 Standard

**employedBy**

**Q:** Silvestre Paredes # employer
**A:** Universidad Politécnica de Cartagena
**Q:** Masashi Kamogawa # employer
**A:** Waseda University
**Q:** Ana Rosa Rama Ballesteros # employer
**A:** University of Jaén # University of Jaén
**Q:** Sara Akbar # employer
**A:** Kuwait Oil Company
**Q:** Pius V # employer
**A:** University of Bologna # University of Pavia
**Q:** Masao Kotani # employer
**A:** University of Tokyo # Osaka University
**Q:** Bernadeta Patro Golab # employer
**A:** Medical University of Warsaw
**Q:** Andy Hertzfeld # employer
**A:** Google

**developedBy**

**Q:** Chronicles of Mystery: The Scorpio Ritual # developer
**A:** CI Games S.A.
**Q:** Call of Duty 3 # developer
**A:** Treyarch # Exakt Entertainment
**Q:** Samplitude # developer
**A:** Bellevue Investments
**Q:** Dangun Feveron # developer
**A:** CAVE
**Q:** Sega Classics Arcade Collection # developer
**A:** Sega
**Q:** Python # developer
**A:** Python Software Foundation # Guido van Rossum
**Q:** Allen Coral Atlas # developer
**A:** Vulcan Inc. # University of Queensland # Carnegie Institution for Science
**Q:** Avatar: The Last Airbender – Into the Inferno # developer
**A:** Nickelodeon

**nativeLanguage**

**Q:** Bill Byrge # native language
**A:** English
**Q:** Augustin Michel # native language
**A:** French
**Q:** Ingrian Finns # native language
**A:** Finnish # Russian
**Q:** Nanu Ram Kumawat # native language
**A:** Rajasthani # Hindi
**Q:** Jumber Dochviri # native language
**A:** Georgian
**Q:** Rosa Estaràs #native language
**A:** Spanish # Catalan
**Q:** Vladimir Fotievich Kozlov # native language
**A:** Russian
**Q:** Vladimir Nemkin # native language
**A:** Russian # Ukrainian

### B.2 Textual context

**inContinent**

**Q:** Reventador # continent
**C:** Daily explosions, ash plumes, lava flows, and incandescent block avalanches during February-July 2022. Volcán El Reventador is located 100 km E of the main... Reventador is an active stratovolcano which lies in the eastern Andes of Ecuador. It lies in a remote area of the national park of the same name, which is...
**A:** South America # Americas

**Q:** Fatimid Caliphate # continent
**C:** The Fatimid Caliphate was an Ismaili Shi'a caliphate extant from the tenth to the twelfth ... Fatimid Caliphate is located in Continental Asia. Encompassing the vast Sahara in North Africa, alongside the Levant in the Middle East, the entirety of the Caliphate consists of Arid biome, save for Sicily and...
**A:** Asia # Africa # Europe

**Q:** Cerro Tenán # continent
**C:** See photos, floor plans and more details about 262 S Paseo Cerro in Green Valley, Arizona. Visit Rent. now for rental rates and other information about this... 200 N Continental Blvd ... NEC Market St & Via Cerro ... Investment Services · Landlord Representation · Tenant Representation · Industrial: Warehouse &...
**A:** Americas

**Q:** Mississippi River # continent
**C:** Feb 10, 2022 ... The Mississippi River is the second longest river in North America, flowing 2,350 miles from its source at Lake Itasca through the center of the... ... or about one-eighth of the entire continent. The Mississippi River lies entirely within the United States. Rising in Lake Itasca in Minnesota, it flows...
**A:** Americas # North America

**Q:** St. Lawrence River # continent
**C:** The St. Lawrence River is a large river in the middle latitudes of North America. Its headwaters begin flowing from Lake Ontario in a roughly northeasterly... St. Lawrence River, hydrographic system of east-central North America. It starts at the outflow of Lake Ontario and leads into the Atlantic Ocean in the...
**A:** North America

**Q:** Cerro El Charabón # continent
**C:** 65, Estancia El Charabón. 49. 66, Área costero-marina Cerro Verde e Islas de la Coronilla-Área General. 48. 67, Area protegida Laguna de Castillos – Tramo... Casa del Sol Boutique Hotel. A cozy stay awaits you in Machu Picchu. ... Altiplánico San Pedro de Atacama ... Welcome to El Charabon. El Charabon.
**A:** Americas

**Q:** Hinterer Seekopf # continent
**C:** Following the breakup of Pangea during the Mesozoic era, the continents of ... of the best day hikes in Kalkalpen National Park is the Hoher Nock – Seekopf. Dec 5, 2016 ... Hinterer Steinbach. Inhaltsverzeichnis aufklappen ... Inhaltsverzeichnis einklappen ... Charakteristik. Hinweise; Subjektive Bewertung...
**A:** Europe

**Q:** Šembera # continent
**C:** Rephrasing Heidegger: A Companion to Heidegger's Being and Time [Sembera, ... Being and Time (Suny Series in Contemporary Continental Philosophy). Feb 26, 2016 ... Coming from Uganda, UNV PO Flavia Sembera was familiar with diversity. ... shared across the continent while experiencing Zambia's beautiful...
**A:** Europe

## work location

**Q:** Karel Lodewijk Sohl # work location
**C:** Karel Lodewijk SOHL geboren op 18 februari 1895 te Maastricht. Hij huwde Johanna Catharina Maria VAN BINSBERGEN 20 november 1924 te Roermond. Karel LANOO. Karen ANDERSON. Karina MROß. Karine LALIEUX ... Lena SOHL. Leonidas MAKRIS. Leopold SPECHT ... Lodewijk ASSCHER. Lora LYUBENOVA. Loren LANDAU ...
**A:** Maastricht # Paris # Maastricht
**Q:** H.G. van Broekhuisen # work location
**C:** Nov 28, 2018 ... Fourth, we are grateful for the work of the e-Vita platform helpdesks: Mireille ... Van Spall HG, Rahman T, Mytton O, Ramasundarahettige C,... Apr 30, 2013 ... Saskia F van Vugt, general practitioner 1,; Berna D L Broekhuizen, ... In Flanders (Belgium) this work was supported by the Research...
**A:** Makassar
**Q:** Hermann Kretzschmer # work location
**C:** The author died in 1890, so this work is in the public domain in its country of origin and other countries and areas where the copyright term is the author's... Stay up to date with Hermann Kretzschmer (German, 1811 – 1890) . Discover works for sale, auction results, market data, news and exhibitions on MutualArt.
**A:** Berlin # Düsseldorf
**Q:** Ole Olsen Teslien # work location
**C:** og Bjom Pedersen Teslien 1831 – kjøpesum 3500 spd. Fra 1845 var Bjom Ptdersen alene eier. ... Efter Ole Jensen overtok sønnen Anders Olsen g. i 1850. Jul 11, 2015 ... Ole Olsen Bak eller Ormhaug som i 1803 kjøpte glrden ØvreViken 461, ... Julius Sand. fra 1819 var Ole Olsen Teslien. someide Nedre n s li,...
**A:** Oslo
**Q:** John Sewell # work location
**C:** John W Sewell was born in 1867 in Elbert County, Georgia, and moved with his parents to Florida when he was 19 years old. Sewell, working for Henry Flagler,... ... Implementation Consulting | Learn more about John Sewell, CMRP's work experience, education, connections & more by visiting their profile on LinkedIn.
**A:** London # London
**Q:** Alan R. Battersby # work location
**C:** Sir Alan Rushton Battersby FRS (4 March 1925 – 10 February 2018) was an English organic chemist best known for his work to define the chemical intermediates... Dec 9, 2018 ... Battersby. Alan R. Battersby, University of Cambridge Cambridge, United Kingdom. "for their fundamental contributions to the elucidation of the...
**A:** Cambridge
**Q:** A. Kate Miller # work location
**C:** Title/Position. Advocacy Director. Department. Advocacy Department. Pronouns. she ... Kate works in a number of issue areas and is always seeking common ground... ... Miller's work experience, education, connections & more by visiting their profile on LinkedIn. ... Work location: Chicago, Illinois, United States. Work...
**A:** Indianapolis
**Q:** Andrzej Grzesik # work location
**C:** View Andrzej Grzesik's profile on LinkedIn, the world's

largest professional community.
Andrzej has 15 jobs listed on
their profile.  Nov 14, 2016 ...
Congratulations to the newest
Java Champion Andrzej Grzesik!
...  in Poland (sfi.org.pl) and
in his work as a Sun Campus Am-
bassador.
**A:** Warsaw

### B.3 Don't Know

**producedBy**

**Q:** Strati # manufacturer
**A:** Local Motors # Oak Ridge Na-
tional Laboratory
**Q:** Philips VG-8235 # manufac-
turer
**A:** Don't know v SS America (1939)
# manufacturer
**A:** Newport News Shipbuilding
**Q:** HMCS Brandon # manufacturer
**A:** Don't know
**Q:** Cluster Platform 3000 SL160z,
Xeon L55xx 2.26 GHz, GigE # manu-
facturer
**A:** Hewlett Packard Enterprise
**Q:** POWER CHALLENGE # manufac-
turer
**A:** Don't know
**Q:** Suzuki Katana # manufacturer
**A:** Suzuki
**Q:** German submarine U-1223 # man-
ufacturer
**A:**Don't know

**spokenLanguage**

**Q:** Allen G. Thurman # languages
spoken, written or signed
**A:** English
**Q:** Rifat Hairy # languages spo-
ken, written or signed
**A:** Don't know
**Q:** Izabela Filipiak # languages
spoken, written or signed
**A:** American English # Polish
**Q:** Vasudev Gopal Paranjpe # lan-
guages spoken, written or signed
**A:** Don't know
**Q:** Jonathan M. Katz # languages
spoken, written or signed

**A:** English
**Q:** Ingeborg Heintze # languages
spoken, written or signed
**A:** Don't know
**Q:** Alicia Coduras Martínez # lan-
guages spoken, written or signed
**A:** Catalan # Spanish
**Q:** Adam Budak # languages spoken,
written or signed
**A:** Don't know