# OpenReview forum: "Evaluating the Knowledge Base Completion Potential of GPT"
_EMNLP/2023/Conference — EMNLP 2023 Findings_

### Official Review · Reviewer_Fktf · 2023-08-02

**Soundness:** 3

**Excitement:**

3: Ambivalent: It has merits (e.g., it reports state-of-the-art results, the idea is nice), but there are key weaknesses (e.g., it describes incremental work), and it can significantly benefit from another round of revision. However, I won't object to accepting it if my co-reviewers champion it.

**Paper Topic And Main Contributions:**

This paper presents an experiment for knowledge base completion by means of GPT-3.
The proposed experiment focuses on the prediction of 41 relations from Petroni et al. (2019). Different promptings have been tested, including a few-shot one to produce a ‘don’t know’ answer. The evaluation has been performed automatically and manually and shows some interesting results on the topic.

**Reasons To Accept:**

The article deals with a topic of certain and current interest on how to use large language models for the construction and expansion of graphs.

**Reasons To Reject:**

I do not find the paper informative enough. For instance, some methodological choices are not well-motivated or explained (apart from the reference) and some results need further testing or explanation (e.g., the lowest scores reached for some of the relations).

**Reproducibility:**

2: Would be hard pressed to reproduce the results. The contribution depends on data that are simply not available outside the author's institution or consortium; not enough details are provided.

**Reviewer Confidence:**

4: Quite sure. I tried to check the important points carefully. It's unlikely, though conceivable, that I missed something that should affect my ratings.

**Typos Grammar Style And Presentation Improvements:**

Line 054, please use \textcite
Line 130, please use \textcite
Line 138, please, check the missing reference
Line 141, The → the
Line 199, please use \textcite
Line 206, please use \textcite
Check the reference format

---

> ### Author Rebuttal · Authors · 2023-08-29
>
> Thank you for your thoughtful comments. If the paper gets accepted, we would use the additional 5th page to expand on the methodological choices and result discussion.
>
> Clarification of experimental setup and reproducibility:
>
> We use a random split of the publicly available dataset WD-Known [1]. We will publish full details as well as Mturk-based human judgements upon acceptance. For reviewer access, we have created a preliminary, anonymous repository at https://drive.google.com/drive/folders/1LNv6k7-TzoqzZfgsHxoqbC6m-jV75USW?usp=sharing
>
> [1] Veseli et al. : Evaluating Language Models for Knowledge Base Completion. European Semantic Web Conference (ESWC) 2023, data repository: https://github.com/bveseli/LMsForKBC

---

### Official Review · Reviewer_5ndf · 2023-08-03

**Typos Grammar Style And Presentation Improvements:** There is a missing reference in line 138
**Soundness:** 4

**Excitement:**

4: Strong: This paper deepens the understanding of some phenomenon or lowers the barriers to an existing research direction.

**Paper Topic And Main Contributions:**

This is a study about the use of GPT-3 for KB completition. The novelty of this study is the fact of paying attention to non-popular statements, what allows to extend KBs in contrast to previous studies.

The main contributions are:
- a study about KB completition on new facts instead of facts already in the KB. This is important to really extend KBs.
- evaluate recall at different precision levels

The paper is well written and easy to understand. I only miss experiments with another LLMs more similar to GPT-3 than Bert-Large (the other model used in this paper). For example, the authors could test an smaller  GPT-3 and study results.  Besides, I guess that the results depends not only on the size of the model, but also on the pre-training documents. I think the authors should study this.

**Reasons To Accept:**

The paper is well written and easy to understand. The authors clearly justify their study based on previous studies that do not pay attention to the long-tail when extending KBs.

The authors also provides a study per relation and using a different prompting.

**Reasons To Reject:**

I only miss experiments with another LLMs more similar to GPT-3 than Bert-Large (the other model used in this paper). For example, the authors could test an smaller  GPT-3 and study results.  Besides, I guess that the results depends not only on the size of the model, but also on the pre-training documents. I think the authors should study this.

**Reproducibility:**

4: Could mostly reproduce the results, but there may be some variation because of sample variance or minor variations in their interpretation of the protocol or method.

**Reviewer Confidence:**

5: Positive that my evaluation is correct. I read the paper very carefully and I am very familiar with related work.

---

> ### Author Rebuttal · Authors · 2023-08-29
>
> Thank you for your thoughtful comments.
>
> Reply to questions:
>
> 1. We performed additional experiments using i) the smaller GPT-3 text-curie-001 with 6B parameters, and ii) GPT-4. As expected, recall@precision90 drops drastically for the smaller GPT-3 model (e.g., from 0.76 to 0.39 for writtenIn, from 0.39 to 0.16 for ownedBy etc.). For GPT-4, the proper evaluation is difficult as its API does not yield confidence scores. Instead, we simply evaluate the overall precision per relation. Some relations are handled very well (e.g., 0.85 for writtenIn, 0.74 for ownedBy), but others still show poor results (e.g., 0.31 for foundedIn, 0.29 for playsInstrument), and the overall recall varies between 30 and 80 percent. The table below shows full results. If accepted, we will include these latest results in the paper.
> 2. The performance does depend on the size and quality of the pre-training corpus. As for the influence of size, there is also prior/parallel work; see, for example:
> N. Kandpal et al.:  Large language models struggle to learn long-tail knowledge. ICML, July 2023.
> Evaluating the influence of the pre-training content quality poses major difficulties, though, as the most interesting GPT models do not convey specifics of their pre-training data. We plan to pursue studies with open-data models such as BLOOM. This is future work.
> 3. Reproducibility:
> We use a random split of the publicly available dataset WD-Known [1]. We will publish full details as well as Mturk-based human judgements upon acceptance. For reviewer access, we have created a preliminary, anonymous repository at https://drive.google.com/drive/folders/1LNv6k7-TzoqzZfgsHxoqbC6m-jV75USW?usp=sharing
>
>     [1] Veseli et al. : Evaluating Language Models for Knowledge Base Completion. European Semantic Web Conference (ESWC) 2023, data repository: https://github.com/bveseli/LMsForKBC
>
> | Relation        | GPT-3 davinci $R@P95$ | GPT-3 davinci $R@P90$ | GPT-3 curie $R@P95$ | GPT-3 curie $R@P90$ | BERT$_{Large}$ $R@P95$ | BERT$_{Large}$ $R@P90$ | GPT-4 $Precision$ | GPT-4 $Recall$ | GPT-3 davinci $Precision$ | GPT-3 davinci $Recall$ |
> |-----------------|---------------------------------|---------------------------------|---------------------------------|---------------------------------|-------------------------|-------------------------|------------|------------|------------------------------|------------------------------|
> | writtenIn       | 0.69                            | 0.76                            | 0.2                             | 0.39                            | 0                       | 0                       | 0.85       | 0.81       | 0.91                         | 0.78                         |
> | ownedBy         | 0.37                            | 0.39                            | 0.11                            | 0.16                            | 0                       | 0                       | 0.74       | 0.54       | 0.6                          | 0.44                         |
> | nativeLanguage  | 0.22                            | 0.7                             | 0.11                            | 0.6                             | 0.43                    | 0.58                    | 0.87       | 0.87       | 0.69                         | 0.86                         |
> | LanguageOfFilm  | 0.21                            | 0.33                            | 0                               | 0.01                            | 0                       | 0.01                    | 0.78       | 0.63       | 0.58                         | 0.52                         |
> | hasCapital      | 0.19                            | 0.31                            | 0                               | 0                               | 0.02                    | 0.02                    | 0.82       | 0.52       | 0.77                         | 0.44                         |
> | officialLanguage | 0.09                            | 0.24                            | 0                               | 0                               | 0.03                    | 0.25                    | 0.69       | 0.68       | 0.67                         | 0.64                         |
> | foundedIn       | 0.06                            | 0.08                            | 0                               | 0                               | 0                       | 0                       | 0.31       | 0.6        | 0.4                          | 0.38                         |
> | playsInstrument | 0.02                            | 0.02                            | 0                               | 0                               | 0                       | 0                       | 0.29       | 0.41       | 0.16                         | 0.18                         |
> | partOf          | 0.01                            | 0.01                            | 0                               | 0                               | 0                       | 0                       | 0.3        | 0.58       | 0.17                         | 0.1                          |
> | citizenOf       | 0.01                            | 0.24                            | 0                               | 0                               | 0.02                    | 0.03                    | 0.75       | 0.65       | 0.67                         | 0.6                          |
> | spokenLanguage  | 0.01                            | 0.47                            | 0.02                            | 0.13                            | 0                       | 0.24                    | 0.57       | 0.73       | 0.54                         | 0.76                         |
> | playerPosition  | 0.01                            | 0.02                            | 0                               | 0                               | 0                       | 0                       | 0.45       | 0.45       | 0.18                         | 0.24                         |
> | inContinent     | 0.01                            | 0.01                            | 0                               | 0                               | 0                       | 0                       | 0.8        | 0.72       | 0.61                         | 0.6                          |
> | namedAfter      | 0.01                            | 0.07                            | 0.01                            | 0.01                            | 0                       | 0                       | 0.68       | 0.67       | 0.53                         | 0.44                         |
> | hostCountry     | 0.01                            | 0.34                            | 0.02                            | 0.02                            | 0                       | 0                       | 0.83       | 0.54       | 0.75                         | 0.48                         |
> | musicLabel      | 0.01                            | 0.02                            | 0                               | 0                               | 0                       | 0                       | 0.65       | 0.68       | 0.16                         | 0.18                         |
> | hasReligion     | 0                               | 0.02                            | 0                               | 0                               | 0.03                    | 0.07                    | 0.54       | 0.82       | 0.47                         | 0.38                         |
> | developedBy     | 0                               | 0.11                            | 0                               | 0                               | 0.04                    | 0.04                    | 0.51       | 0.83       | 0.45                         | 0.5                          |
> | countryOfJurisdiction | 0                        | 0.08                            | 0.05                            | 0.08                            | 0.02                    | 0.02                    | 0.37       | 0.23       | 0.52                         | 0.22                         |
> | subclassOf      | 0                               | 0.37                            | 0                               | 0                               | 0                       | 0                       | 0.16       | 0.31       | 0.73                         | 0.85                         |
> | diplomaticRelation | 0                             | 0.01                            | 0                               | 0                               | 0                       | 0                       | 0.41       | 0.7        | 0.7                          | 0.17                         |
> | CountryOfOrigin | 0                               | 0.01                            | 0                               | 0                               | 0.01                    | 0.01                    | 0.66       | 0.38       | 0.48                         | 0.31                         |
>
> **Table**: In this table, GPT-3 curie refers to GPT-3 text-curie-001, and GPT-3 davinci refers to GPT-3 text-davinci-003.

---

### Official Review · Reviewer_QRC9 · 2023-08-04

**Soundness:** 4

**Excitement:**

4: Strong: This paper deepens the understanding of some phenomenon or lowers the barriers to an existing research direction.

**Paper Topic And Main Contributions:**

The paper presents an evaluation of GPT-3 on the potential of its use for KB completion. They used a recent KB completion benchmark dataset to prompt GPT-3 to get the relations generated. The work is more of an evaluation setup to see how good GPT-3 can be for the KB completion task. I think this analysis may provide good insights to works around KGs and LLMs, especially LLMs’ capabilities on knowledge related tasks.

**Questions For The Authors:**

Question A: It is not clear to me how someone would use GPT-3 to get high precision outcome. Is it the case that we can prompt GPT-3 and based on a predefined threshold to filter out predictions so that we may get roughly high precision results as mentioned in this paper? If so, make it clear in the paper.

Question B: Then, do we have to define a threshold for each and every relation in the KG?


**Reasons To Accept:**

•	Paper is easy to understand and provides a timely analysis of GPT-3 capabilities on KG completion task (how useful it would be)

•	Findings presented in the paper may be of interest to a broader community.

**Reasons To Reject:**

•	It is not clear how someone would use the insights presented in the paper to get high precision results (also see questions below).

**Reproducibility:**

4: Could mostly reproduce the results, but there may be some variation because of sample variance or minor variations in their interpretation of the protocol or method.

**Reviewer Confidence:**

4: Quite sure. I tried to check the important points carefully. It's unlikely, though conceivable, that I missed something that should affect my ratings.

---

> ### Author Rebuttal · Authors · 2023-08-29
>
> Thank you for your thoughtful comments.
>
> Reply to questions:
>
> A. If the goal solely is to achieve high precision, we would sort GPT-3 outputs by confidence scores with cut-off at some threshold. However, there is no easy way to aim for maximal recall at sufficiently high precision. The point of our paper is that this realistic goal for KB completion poses a major challenge.
>
> B. Yes, such thresholds should be specifically chosen for each relation. Using withheld ground-truth data, we can automatically compute these per-relation thresholds such that the model yields a desired target precision (presumably at limited recall, though).

---

### Meta-Review · Area_Chair_9NYi · 2023-09-19

**Recommendation:** 4

**Metareview:**

The paper investigated the potential of using LLMs (GPT-3) for KB completion. The paper is easy to understand and provides a timely analysis of LLMs' capabilities on KG completion tasks. Findings presented in the paper may be of interest to a broader community. The authors clearly justify their study based on previous studies that do not pay attention to the long tail when extending KBs. The authors also provide a study per relation and use a different prompting. Overall, it is a solid short paper presenting interesting findings.

---

### Decision · Program_Chairs · 2023-10-07

**Decision:**

Accept-Findings

**Comment:**

The paper investigated the potential of using LLMs (GPT-3) for KB completion. The paper is easy to understand and provides a timely analysis of LLMs' capabilities on KG completion tasks. Findings presented in the paper may be of interest to a broader community. The authors clearly justify their study based on previous studies that do not pay attention to the long tail when extending KBs. The authors also provide a study per relation and use a different prompting. Overall, it is a solid short paper presenting interesting findings.